# VAE-Inf: A statistically interpretable generative paradigm for imbalanced classification

## Abstract

Imbalanced classification remains a pervasive challenge in machine learning, particularly when minority samples are too scarce to provide a robust discriminative boundary. In such extreme scenarios, conventional models often suffer from unstable decision boundaries and a lack of reliable error control. To bridge the gap between generative modeling and discriminative classification, we propose a two-stage framework **VAE-Inf** that integrates deep representation learning with statistically interpretable hypothesis testing. In the first stage, we adopt a one-class modeling perspective by training a variational autoencoder (VAE) exclusively on majority-class data to capture the underlying reference distribution. The resulting latent posteriors are aggregated via a Wasserstein barycenter to construct a global Gaussian reference model, providing a geometrically principled baseline for the majority class. In the second stage, we transform this generative foundation into a discriminative classifier by fine-tuning the encoder with limited minority samples. This is achieved through a novel distribution-aware loss that enforces probabilistic separation between classes based on variance-normalized projection statistics. For inference, we introduce a projection-based score that admits a natural hypothesis testing interpretation, allowing for a distribution-free calibration procedure. This approach yields exact finite-sample control of the Type-I error (false positive rate) without relying on restrictive parametric assumptions. Extensive experiments on diverse real-world benchmarks demonstrate that our framework achieves competitive performance against other approaches. The codes are available upon request.

## 1 Introduction

Imbalanced classification is a fundamental challenge in machine learning and statistics, where the majority class contains far more samples than the minority class. This phenomenon commonly arises in real-world applications such as pattern recognition (Dong et al., 2019), object detection (Oksuz et al., 2021) and medical diagnosis (Guo et al., 2017). In these domains, correct identification of minority samples is essential, while misclassifying them can be costly or even harmful. For instance, in disease screening, missing even a few true positive cases may lead to severe medical consequences or even loss of life. However, standard learning methods typically focus on maximizing overall accuracy (Pei et al., 2024) and assume equal importance among all samples, which causes the learned model to be dominated by the majority class and perform poorly in rare but crucial cases.

Many strategies have been developed to address class imbalance in traditional learning frameworks. Existing approaches can generally be categorized into three groups: (1) Resampling methods focus on resampling the training data to rebalance class proportions (He & Garcia, 2009). Typical approaches include random under-sampling of the majority class (Tahir et al., 2009) and over-sampling of the minority class, as well as synthetic data generation techniques such as the Synthetic Minority Over-sampling Technique (SMOTE) and its variants (Chawla et al., 2002; Zheng et al., 2016). (2) Cost-sensitive learning methods (Elkan, 2001; Thai-Nghe et al., 2010; Castro & Braga, 2013) modify the loss function to impose higher penalties on misclassified minority samples. This allows the model to pay greater attention to rare examples. (3) Ensemble learning methods combine multiple classifiers or resampled subsets to improve performance (Galar et al.,

2012). Representative ensemble-based algorithms include AdaBoost (Freund & Schapire, 1996), SMOTE-Boost (Chawla et al., 2003), and Bagging (Galar et al., 2013), which integrate sampling or reweighting strategies within the ensemble framework. However, these traditional approaches may struggle to effectively capture complex structures in high-dimensional data (Huang et al., 2025).

Recently, deep learning has achieved remarkable success due to its exceptional learning capacity, and a growing number of methods have applied deep architectures to solve imbalance classification problems. In particular, deep generative models are integrated to offer high-quality synthetic samples for the minority class (Mullick et al., 2019; Wang et al., 2020). Generative Adversarial Networks (GANs) (Goodfellow et al., 2014) have been widely used as oversampling tools for image data (Sampath et al., 2021), with variants such as BAGAN (Mariani et al., 2018) and CGAN (Nazari & Branco, 2021) designed to improve generation quality. Similarly, Variational Autoencoders (VAEs) (Kingma & Welling, 2013) have also been adopted to generate minority samples in a probabilistic latent space (Wan et al., 2017; Zhang et al., 2018). Moreover, new loss functions, such as Focal loss (Lin et al., 2017) and Label-Distribution-Awareness margin (LDAM) (Cao et al., 2019), have been proposed to replace the commonly used Cross Entropy (CE) loss to improve discrimination. Specifically, the Focal loss reduces the weight of well-classified samples to focus training on difficult instances, while the LDAM loss explicitly adjusts the classification margin according to the frequency of the class.

Despite the progress of deep learning-based approaches, extreme class imbalance remains a fundamental challenge: the minority class is often too sparse to faithfully characterize its underlying distribution. While data-level generation can alleviate sample scarcity, its effectiveness is strictly capped by the fidelity of synthetic examples due to the very limited available training samples of the minority class. For instance, GAN-based oversampling frequently suffers from mode collapse (Arjovsky et al., 2017), while existing latent-space models often rely on restrictive symmetric assumptions (Guo et al., 2019) that fail to capture the complex, asymmetric feature structures of real-world data. These limitations underscore a critical insight: in extreme scenarios, defining what the minority class "is" is significantly harder than defining what it "is not".

Building on this insight, we reframe extreme imbalanced classification through the lens of statistical inference modeling, drawing inspiration from anomaly detection (Ruff et al., 2018; Pang et al., 2019; Ruff et al., 2020; Zhou et al., 2022). Instead of constructing fragile models for a sparse minority class, we focus on establishing a robust profile of the majority class (normality) and identifying minority samples as significant deviations from this regularity. In particular, we propose a two-stage framework VAE-Inf that integrates deep representation learning with statistical inference. In Stage 1, a VAE is pretrained exclusively on majority-class data to learn a continuous probabilistic latent manifold. In Stage 2, the encoder is fine-tuned using limited minority samples through a distribution-aware objective, which enhances class separability while preserving the learned majority structure. This design enables a statistically interpretable inference strategy, allowing for decision rules with controllable error behavior and finite-sample guarantees. Our numerical results demonstrate that even a small amount of labeled minority information can be utilized to improve the detection performance of the minority class, making it particularly suitable for imbalanced tasks. The main contributions of this paper are summarized as follows:

- We propose a two-stage VAE-based framework VAE-Inf that constructs a reference distribution of the majority class and effectively utilizes limited minority samples through a distribution-aware fine-tuning process.

- We develop a projection-score calibration mechanism that provides finite-sample Type-I error control under exchangeability, while empirically achieving competitive Type-II trade-offs.

- Extensive experiment results on several important datasets demonstrate that our VAE-Inf achieves superior classification performance while maintaining reliable Type-I/II error control compared with imbalanced learning and anomaly detection baselines.

The remainder of this paper is organized as follows. We start with the notation and problem formulation. Section 2 presents the proposed method. Section 3 reports the experimental setup and results, and Section 4 concludes the paper.

## 1.1 Notation and Problem Formulation

Formally, consider a binary classification problem with input–label pairs $(\boldsymbol{x}, y)$, where $\boldsymbol{x} \in \mathbb{X} \subseteq \mathbb{R}^p$ and $y \in \{1, 2\}$. Assign $y = 1$ to the majority class and $y = 2$ to the minority class. We formulate the problem from a hypothesis testing perspective. Given an observation $\boldsymbol{x}$, we consider the hypotheses

$$H_0 : \boldsymbol{x} \sim P_1 \quad \text{vs.} \quad H_1 : \boldsymbol{x} \sim P_2, \tag{1}$$

where $P_1$ and $P_2$ denote the data distributions of the majority and minority classes, respectively. A decision rule $\psi : \mathbb{X} \to \{1, 2\}$ assigns each sample to one of the two classes. Type-I and Type-II errors are defined as $R_1(\psi) = \mathbb{P}_{\boldsymbol{x} \sim P_1}\big(\psi(\boldsymbol{x}) = 2\big)$ and $R_2(\psi) = \mathbb{P}_{\boldsymbol{x} \sim P_2}\big(\psi(\boldsymbol{x}) = 1\big)$. In general, $R_1(\psi)$ and $R_2(\psi)$ can not be minimized simultaneously, leading to an inherent trade-off between false positive and false negative rates.

We adopt a single-error control perspective that prioritizes reliable regulation of one error type. Specifically, given a user-specified tolerance level $\delta \in (0, 1)$, our goal is to construct a decision rule $\psi$ such that $R_1(\psi) \leq \delta$. This formulation emphasizes controlled statistical decision-making, rather than purely optimizing classification accuracy.

## 2 Methodologies

In this paper, we propose a novel framework VAE-Inf that addresses imbalanced classification by combining majority distribution modeling with statistically controlled decision-making. Our core idea is to model and learn the distribution of the majority class in latent space, and utilize it to perform error-aware classification with limited minority data. As illustrated in Figure 1, the framework consists of two stages. In Stage 1, a VAE is pretrained exclusively on the majority-class samples to learn a latent reference distribution. In Stage 2, the encoder is fine-tuned using a distribution-aware objective that promotes separation between majority and minority representations while preserving the learned reference structure. At inference time, we propose a projection-based statistic in the latent space, which serves as a decision score for hypothesis testing and enables controlled error trade-offs.

## 2.1 Stage 1: Pretraining VAE on the Majority Class

In the first stage, we pretrain a VAE exclusively on majority-class samples, aiming to learn a latent representation of the normal data manifold. This stage serves to construct a statistical reference that characterizes the majority distribution in latent space.

**VAE Training.** Let $\mathcal{D}_1 = \{\boldsymbol{x}_i\}_{i=1}^{N_1}$ denote the set of majority-class samples with $\boldsymbol{x}_i \in \mathbb{R}^d$. We employ a standard VAE with encoder $q_\phi(\boldsymbol{z}|\boldsymbol{x})$ and decoder $p_\theta(\boldsymbol{x}|\boldsymbol{z})$, where the encoder defines a diagonal Gaussian posterior $q_\phi(\boldsymbol{z}|\boldsymbol{x}) = \mathcal{N}\big(\boldsymbol{z}; \boldsymbol{\mu}_\phi(\boldsymbol{x}), \text{diag}(\boldsymbol{\sigma}_\phi^2(\boldsymbol{x}))\big)$. Latent variables are sampled via the reparameterization trick:

$$\boldsymbol{z} = \boldsymbol{\mu}_\phi(\boldsymbol{x}) + \boldsymbol{\sigma}_\phi(\boldsymbol{x}) \odot \boldsymbol{\epsilon}, \quad \boldsymbol{\epsilon} \sim \mathcal{N}(\boldsymbol{0}, \boldsymbol{I}), \tag{2}$$

where $\odot$ is the element-wise product. The model is trained by maximizing the evidence lower bound (ELBO) (Kingma & Welling, 2013):

$$\mathcal{L}_{\text{VAE}}(\boldsymbol{x}) = \mathbb{E}_{\boldsymbol{z} \sim q_\phi(\boldsymbol{z}|\boldsymbol{x})} \left[\log p_\theta(\boldsymbol{x}|\boldsymbol{z})\right] - D_{\text{KL}}(q_\phi(\boldsymbol{z}|\boldsymbol{x}) \,\|\, p(\boldsymbol{z})),$$

where $p(\boldsymbol{z}) = \mathcal{N}(0, \boldsymbol{I})$ is the standard Gaussian prior. This training procedure yields a collection of posterior distributions $\{q_\phi(\boldsymbol{z}|\boldsymbol{x}_i)\}_{i=1}^{N_1}$, providing local probabilistic descriptions of majority samples in latent space.

**Estimates of Global Mean and Variance** While individual posteriors $q_\phi(\boldsymbol{z}|\boldsymbol{x})$ provide local descriptions of each sample, we aim to construct a global characterization of the majority class in latent space. To this end, we aggregate the collection of Gaussian posteriors $\{\mathcal{N}(\boldsymbol{z}; \boldsymbol{\mu}_{\boldsymbol{x}_i}, \text{diag}(\boldsymbol{\sigma}_\phi^2(\boldsymbol{x}_i)))\}_{i=1}^{N_1}$ into a single latent reference distribution. We define this reference as the 2-Wasserstein barycenter of the individual posteriors:

$$\mathcal{N}(\boldsymbol{z}; \boldsymbol{\mu}_{\text{ref}}, \boldsymbol{\Sigma}_{\text{ref}}) = \arg\min_{\mathcal{N}(\boldsymbol{z}; \boldsymbol{\mu}, \boldsymbol{\Sigma})} \sum_{i=1}^{N_1} W_2^2\big(\mathcal{N}(\boldsymbol{z}; \boldsymbol{\mu}_{\boldsymbol{x}_i}, \text{diag}(\boldsymbol{\sigma}_\phi^2(\boldsymbol{x}_i))), \, \mathcal{N}(\boldsymbol{z}; \boldsymbol{\mu}, \boldsymbol{\Sigma})\big),$$

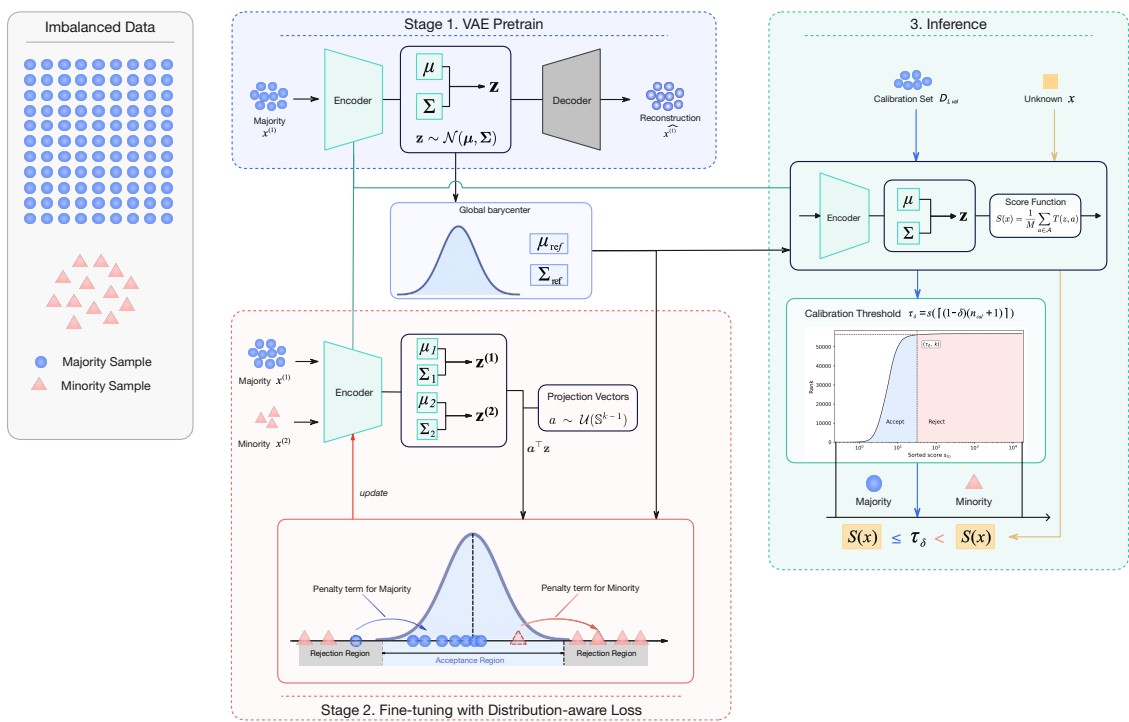

Figure 1: The overview of the proposed two-stage VAE-Inf for imbalanced classification. (1) Stage 1 learns a latent reference distribution of the majority class in latent space using a VAE. (2) Stage 2 refines the representation via a distribution-aware objective to enhance separation between majority and minority samples. (3) At inference, a projection-based score is computed with respect to the learned reference distribution, enabling statistically principled decision-making with controlled error trade-offs.

where $\boldsymbol{\mu}_{\boldsymbol{x}_i} \in \mathbb{R}^k$ and $\mathrm{diag}(\boldsymbol{\sigma}_\phi^2(\boldsymbol{x}_i)) \in \mathbb{R}^{k \times k}$ denote the encoder-predicted posterior mean and corresponding diagonal covariance for each sample $\boldsymbol{x}_i \in \mathcal{D}_1$. For two Gaussians $\mathcal{N}(\boldsymbol{z}; \boldsymbol{\mu}_1, \boldsymbol{\Sigma}_1)$ and $\mathcal{N}(\boldsymbol{z}; \boldsymbol{\mu}_2, \boldsymbol{\Sigma}_2)$ with diagonal covariances, the 2-Wasserstein distance reduces to

$$W_2^2(\mathcal{N}(\boldsymbol{z}; \boldsymbol{\mu}_1, \boldsymbol{\Sigma}_1), \, \mathcal{N}(\boldsymbol{z}; \boldsymbol{\mu}_2, \boldsymbol{\Sigma}_2)) = \|\boldsymbol{\mu}_1 - \boldsymbol{\mu}_2\|_2^2 \, + \, \|\boldsymbol{\Sigma}_1^{1/2} - \boldsymbol{\Sigma}_2^{1/2}\|_F^2,$$

where $\|\cdot\|_F$ is the Frobenius norm. Following Mallasto & Feragen (2017), the Wasserstein barycenter admits closed-form estimates for the mean and diagonal covariance:

$$\boldsymbol{\mu}_{\mathrm{ref}} = \frac{1}{N_1} \sum_{i=1}^{N_1} \boldsymbol{\mu}_{\boldsymbol{x}_i}, \tag{3}$$

$$\boldsymbol{\Sigma}_{\mathrm{ref}} = \mathrm{diag}\left( \left( \frac{1}{N_1} \sum_{i=1}^{N_1} \sqrt{\boldsymbol{\sigma}_\phi^2(\boldsymbol{x}_i)} \right)^2 \right). \tag{4}$$

## 2.2 Stage 2: Fine-tuning with a Distribution-aware Statistical Margin

While Stage 1 provides a reference distribution for the majority class, the resulting latent space is not yet optimized for discrimination between the majority class and minority class. The objective of Stage 2 is therefore to fine-tune the encoder under a distribution-aware statistical criterion: we seek a representation in which majority samples remain compatible with the reference distribution learned in Stage 1, whereas minority samples are systematically pushed away from it.

### 2.2.1 Projection-based Statistics for Hypothesis Testing

It is challenging to conduct hypothesis testing in the latent space due to the high dimensionality of $\mathbf{z} \in \mathbb{R}^k$. To address this, we leverage the projection property of multivariate Gaussians to reduce the high-dimensional comparison problem to a family of one-dimensional statistical comparisons. Under the Stage 1 reference model, $\mathbf{z} \sim \mathcal{N}(\boldsymbol{\mu}_{\text{ref}}, \boldsymbol{\Sigma}_{\text{ref}})$, for any deterministic direction $\boldsymbol{a} \in \mathbb{R}^k$, the projection $\boldsymbol{a}^\top \mathbf{z}$ is univariate Gaussian:

$$\boldsymbol{a}^\top \mathbf{z} \sim \mathcal{N}(\boldsymbol{a}^\top \boldsymbol{\mu}_{\text{ref}}, \, \boldsymbol{a}^\top \boldsymbol{\Sigma}_{\text{ref}} \boldsymbol{a}).$$

To quantify the deviation of a latent sample from the majority reference along direction $\boldsymbol{a}$, we define the squared projected deviation

$$d(\boldsymbol{z}, \boldsymbol{a}) := \left( \boldsymbol{a}^\top \boldsymbol{z} - \boldsymbol{a}^\top \boldsymbol{\mu}_{\text{ref}} \right)^2.$$

Normalizing this quantity by the projected variance yields the projection statistic

$$T(\boldsymbol{z}, \boldsymbol{a}) = \frac{d(\boldsymbol{z}, \boldsymbol{a})}{\boldsymbol{a}^\top \boldsymbol{\Sigma}_{\text{ref}} \boldsymbol{a}} = \frac{\left( \boldsymbol{a}^\top \boldsymbol{z} - \boldsymbol{a}^\top \boldsymbol{\mu}_{\text{ref}} \right)^2}{\boldsymbol{a}^\top \boldsymbol{\Sigma}_{\text{ref}} \boldsymbol{a}}. \tag{5}$$

When $\boldsymbol{z} \sim \mathcal{N}(\boldsymbol{\mu}_{\text{ref}}, \boldsymbol{\Sigma}_{\text{ref}})$ and $\boldsymbol{a}$ is fixed, the standardized projection satisfies

$$\frac{\boldsymbol{a}^\top \boldsymbol{z} - \boldsymbol{a}^\top \boldsymbol{\mu}_{\text{ref}}}{\sqrt{\boldsymbol{a}^\top \boldsymbol{\Sigma}_{\text{ref}} \boldsymbol{a}}} \sim \mathcal{N}(0, 1).$$

Therefore, $T(\boldsymbol{z}, \boldsymbol{a})$ provides a direction-wise test statistic for assessing whether a latent representation is statistically consistent with the majority reference distribution.

### 2.2.2 Distribution-aware Statistical Margin Loss

Let $\boldsymbol{x}^{(1)} \in \mathcal{D}_1$ be a sample from the majority class and $\boldsymbol{x}^{(2)} \in \mathcal{D}_2$ be a sample from the minority class. Their latent representations are sampled from the pretrained encoder posteriors via the reparameterization trick:

$$\boldsymbol{z}^{(i)} = \boldsymbol{\mu}_\phi(\boldsymbol{x}^{(i)}) + \boldsymbol{\sigma}_\phi(\boldsymbol{x}^{(i)}) \odot \boldsymbol{\epsilon}, \qquad \boldsymbol{\epsilon} \sim \mathcal{N}(\boldsymbol{0}, \boldsymbol{I}), \qquad i = 1, 2.$$

Under the Gaussian reference model, $T(\boldsymbol{z}, \boldsymbol{a})$ follows a chi-square($\chi^2$) distribution with one degree of freedom. Therefore, for a prescribed threshold $\alpha > 0$, the event $T(\boldsymbol{z}, \boldsymbol{a}) \leq \alpha$ defines a direction-wise acceptance region with respect to the majority reference distribution. Based on the hypotheses in Eq. (1), this yields a natural testing interpretation: if $T(\boldsymbol{z}, \boldsymbol{a})$ lies in the acceptance region, $H_0$ is not rejected; otherwise, $H_0$ is rejected.

For a fixed projection direction $\boldsymbol{a}$, the desired class separation can be expressed as

$$T(\boldsymbol{z}^{(1)}, \boldsymbol{a}) \leq \alpha, \qquad T(\boldsymbol{z}^{(2)}, \boldsymbol{a}) > \alpha, \tag{6}$$

where $\alpha$ serves as a direction-wise statistical margin. Moreover, $\alpha$ admits a direct probabilistic interpretation: choosing $\alpha = c^2$ corresponds to a $c$-sigma tolerance in the standardized projected space. For example, the critical region $T(\boldsymbol{z}, \boldsymbol{a}) > 9$ corresponds to the $3\sigma$ rule, which rejects $H_0$ with significance level $\approx 0.27\%$.

Since $T(\boldsymbol{z}, \boldsymbol{a})$ is defined as Eq. (5), Eq. (6) can be equivalently written as

$$d(\boldsymbol{z}^{(1)}, \boldsymbol{a}) \leq \alpha \, (\boldsymbol{a}^\top \boldsymbol{\Sigma}_{\text{ref}} \boldsymbol{a}), \qquad d(\boldsymbol{z}^{(2)}, \boldsymbol{a}) > \alpha \, (\boldsymbol{a}^\top \boldsymbol{\Sigma}_{\text{ref}} \boldsymbol{a}).$$

Rather than relying on a single global discrepancy such as the Mahalanobis distance, we sample random directions $\boldsymbol{a} \sim \mathcal{U}(\mathbb{S}^{k-1})$. This allows the model to detect departures from the majority manifold across multiple latent-space orientations and provides diverse gradient signals during training, thereby discouraging overfitting to a single global metric. We therefore design the following distribution-aware regularization loss:

$$\mathcal{L}_{\text{reg}} = \mathbb{E}_{\boldsymbol{x}^{(1)} \sim \mathcal{D}_1, \, \boldsymbol{x}^{(2)} \sim \mathcal{D}_2} \, \mathbb{E}_{\boldsymbol{a} \sim \mathcal{U}(\mathbb{S}^{k-1})} \left[ \left( d(\boldsymbol{z}^{(1)}, \boldsymbol{a}) - \alpha(\boldsymbol{a}^\top \boldsymbol{\Sigma}_{\text{ref}} \boldsymbol{a}) \right)_+ + \beta \left( \alpha(\boldsymbol{a}^\top \boldsymbol{\Sigma}_{\text{ref}} \boldsymbol{a}) - d(\boldsymbol{z}^{(2)}, \boldsymbol{a}) \right)_+ \right], \tag{7}$$

where $(x)_+ := \max(0, x)$ for any $x \in \mathbb{R}$, and $\beta > 0$ controls the relative strength of the minority penalty. The first term is a *majority-tightening* term, which penalizes majority samples whose projected deviations exceed the prescribed high-probability region of the reference model. The second term is a *minority-pushing* term, which penalizes minority samples that remain too close to the majority reference and therefore fail to enter the rejection region.

### 2.3 Inference with Distribution-Free Calibration

The final component of VAE-Inf is a distribution-free calibration procedure that converts latent anomaly scores into statistically valid decision rules. While Stage 2 uses projection-based statistical structure to shape the latent space, the resulting scores are not automatically calibrated for decision-making. We employ an empirical calibration scheme that provides finite-sample control of the Type-I error under exchangeability, without requiring parametric assumptions on the score distribution. In short, exchangeability means that the joint distribution is invariant under permutations of the random variables (Chow & Teicher, 2003).

Let $\mathcal{D} = \mathcal{D}_1 \cup \mathcal{D}_2$ denote the full dataset, where $\mathcal{D}_1$ and $\mathcal{D}_2$ correspond to the majority and minority classes, respectively. $\mathcal{D}_1$ and $\mathcal{D}_2$ are further partitioned as $\mathcal{D}_1 = \mathcal{D}_{1,\text{train}} \cup \mathcal{D}_{1,\text{val}} \cup \mathcal{D}_{1,\text{test}}, \quad \mathcal{D}_2 = \mathcal{D}_{2,\text{train}} \cup \mathcal{D}_{2,\text{val}} \cup \mathcal{D}_{2,\text{test}}$. The majority validation split $\mathcal{D}_{1,\text{val}}$ is used as a held-out calibration set for threshold selection, and we denote $n_{\text{cal}} := |\mathcal{D}_{1,\text{val}}|$. Given an input $\boldsymbol{x}$, the encoder defines $q_\phi(\boldsymbol{z} \mid \boldsymbol{x}) = \mathcal{N}\big(\boldsymbol{z}; \boldsymbol{\mu}_\phi(\boldsymbol{x}), \text{diag}(\boldsymbol{\sigma}_\phi^2(\boldsymbol{x}))\big)$. We obtain a latent code $\boldsymbol{z}$ by sampling from this posterior via the reparameterization trick in Eq. (2). The statistic $T(\boldsymbol{z}, \boldsymbol{a})$ in Eq. (5) measures deviation from the majority reference along the projection direction $\boldsymbol{a}$. To obtain a more stable anomaly score, we aggregate over a set of directions $\mathcal{A} = \{\boldsymbol{a}_1, \dots, \boldsymbol{a}_M\}$ sampled uniformly from the unit sphere, we define

$$S(\boldsymbol{x}) = \frac{1}{M} \sum_{\boldsymbol{a} \in \mathcal{A}} T(\boldsymbol{z}, \boldsymbol{a}), \tag{8}$$

where larger values indicate stronger deviation from the majority distribution. In practice, the projection set $\mathcal{A}$ is fixed after sampling.

We then formulate classification as the one-sided hypothesis test in Eq. (1) using $S(\boldsymbol{x})$ as the test value. To control the Type-I error at a prescribed level $\delta \in (0,1)$, we calibrate the decision threshold using the empirical distribution of calibration scores from the majority class. Specifically, for each $\boldsymbol{x}_i \in \mathcal{D}_{1,\text{val}}$, we compute $S_i = S(\boldsymbol{x}_i)$, and denote the sorted scores by $S_{(1)} \leq \cdots \leq S_{(n_{\text{cal}})}$. The threshold is chosen as the empirical upper quantile, defined by

$$k = \lceil (1 - \delta)(n_{\text{cal}} + 1) \rceil, \qquad \tau_\delta = \begin{cases} S_{(k)}, & k \leq n_{\text{cal}}, \\ +\infty, & k = n_{\text{cal}} + 1. \end{cases} \tag{9}$$

Under the sole assumption that the calibration scores and the score of any future majority sample are exchangeable (Vovk et al., 2005; Lei et al., 2018; Romano et al., 2019). this threshold yields the following finite-sample marginal guarantee. This result is formalized in Theorem 1, with proof deferred to Appendix A.2.

**Theorem 1 (Finite-Sample Type-I Error Control)** *Let $S_1, \dots, S_{n_{\text{cal}}}$ be the scores of the majority-class calibration samples, and let $S_{n_{\text{cal}}+1}$ be the score of a future majority sample. Assume that the scores are exchangeable, i.e., the joint distribution of $(S_1, \dots, S_{n_{\text{cal}}}, S_{n_{\text{cal}}+1})$ is invariant under permutations. Equivalently, for every permutation $\pi$ of $\{1, \dots, n_{\text{cal}} + 1\}$,*

$$(S_1, \dots, S_{n_{\text{cal}}}, S_{n_{\text{cal}}+1}) \stackrel{d}{=} (S_{\pi(1)}, \dots, S_{\pi(n_{\text{cal}}+1)}).$$

*Let $\tau_\delta$ be the empirical upper quantile defined in Eq. (9) from the calibration scores. Then*

$$\mathbb{P}\big(S_{n_{\text{cal}}+1} \leq \tau_\delta\big) \geq 1 - \delta.$$

Theorem 1 guarantees that a future majority score is accepted with probability at least $1 - \delta$. Based on the calibrated threshold $\tau_\delta$, we define the final prediction rule for a test sample $\boldsymbol{x}$ by

$$\psi(\boldsymbol{x}) = \begin{cases} 1, & S(\boldsymbol{x}) \leq \tau_\delta, \\ 2, & S(\boldsymbol{x}) > \tau_\delta, \end{cases}$$

where class 1 denotes the majority class and class 2 denotes the minority class. For a future majority sample $\boldsymbol{x} \sim P_1$, the Type-I error occurs if and only if $S(\boldsymbol{x}) > \tau_\delta$. Therefore,

$$R_1(\psi) = \mathbb{P}_{\boldsymbol{x} \sim P_1}\big(\psi(\boldsymbol{x}) = 2\big) = \mathbb{P}_{\boldsymbol{x} \sim P_1}\big(S(\boldsymbol{x}) > \tau_\delta\big) = 1 - \mathbb{P}_{\boldsymbol{x} \sim P_1}\big(S(\boldsymbol{x}) \leq \tau_\delta\big) \leq \delta,$$

where the last inequality follows from Theorem 1. Hence, the resulting classifier is explicitly calibrated to control the Type-I error at level $\delta$, yielding a transparent and statistically interpretable decision mechanism. In particular, this error-control guarantee is distribution-free and does not depend on the particular modeling choices used to construct the latent score. As a result, the significance level $\delta$ serves as a reliable and operationally meaningful indicator of the Type-I error in practice.

For clarity, the overall training and inference procedures are summarized in Algorithms 1 and 2, respectively.

---

**Algorithm 1** Training algorithm of the proposed two-stage VAE-Inf

---

1: **Input:** $\mathcal{D} = \mathcal{D}_{\text{train}} \cup \mathcal{D}_{\text{val}} \cup \mathcal{D}_{\text{test}}$; $\alpha > 0$; $\beta > 0$; number of projections $M$; training epochs $E_1, E_2$
2: **Initialize:** encoder $q_\phi(z|x)$, decoder $p_\theta(x|z)$, latent dimension $d_l$
3: **Stage 1: VAE Training on Majority Class**
4: **for** epoch = 1 to $E_1$ **do**
5:     Sample the mini-batch $\mathcal{B}_1 \subset \mathcal{D}_{1,\text{train}}$
6:     $(\boldsymbol{\mu}, \boldsymbol{\sigma}^2) \leftarrow q_\phi(\mathcal{B}_1)$                                              ▷ Encoding
7:     $\boldsymbol{\epsilon} \sim \mathcal{N}(\mathbf{0}, \boldsymbol{I})$
8:     $\boldsymbol{z} \leftarrow \boldsymbol{\mu} + \boldsymbol{\sigma} \odot \boldsymbol{\epsilon}$                                      ▷ Reparameterization
9:     Compute $\mathcal{L}_{\text{VAE}}$ and update $(\phi, \theta)$                        ▷ ELBO optimization
10: **end for**
11: Compute $(\boldsymbol{\mu}_{\text{ref}}, \boldsymbol{\Sigma}_{\text{ref}})$ from $\mathcal{D}_{1,\text{train}}$ using Eq. (3) and Eq. (4)     ▷ Wasserstein barycenter (diagonal)
12: **Stage 2: Encoder Fine-tuning with Distribution-aware Loss**
13: Fix decoder parameters $\theta$                                                    ▷ Freeze decoder
14: **for** epoch = 1 to $E_2$ **do**
15:     Sample $\mathcal{A} = \{\boldsymbol{a}_j\}_{j=1}^M$, where $\boldsymbol{a}_j \sim \mathcal{U}(\mathbb{S}^{k-1})$                    ▷ Random projections
16:     Sample $\mathcal{B}_1 \subset \mathcal{D}_{1,\text{train}}$ and $\mathcal{B}_2 \subset \mathcal{D}_{2,\text{train}}$
17:     Draw latent codes $\boldsymbol{z}$ for $\mathcal{B}_1 \cup \mathcal{B}_2$ via reparameterization
18:     Compute $\mathcal{L}_{\text{reg}}$ in Eq. (7) and update $\phi$                   ▷ Distribution-aware separation
19: **end for**
20: **return** $(\phi, \theta)$, $(\boldsymbol{\mu}_{\text{ref}}, \boldsymbol{\Sigma}_{\text{ref}})$

---

**Algorithm 2** Inference with distribution-free calibration

---

1: **Input:** trained $(\phi, \theta)$; $(\boldsymbol{\mu}_{\text{ref}}, \boldsymbol{\Sigma}_{\text{ref}})$; $\mathcal{D}_{1,\text{val}}$; projections $\mathcal{A}$; target Type-I level $\delta$; test sample $\boldsymbol{x}$
2: **Calibration: Empirical Quantile Threshold**
3: Compute calibration scores $S_i \leftarrow S(\boldsymbol{x}_i)$ for all $\boldsymbol{x}_i \in \mathcal{D}_{1,\text{val}}$
4: Sort $\{S_i\}$ increasingly as $S_{(1)} \leq \cdots \leq S_{(n_{\text{cal}})}$
5: $k \leftarrow \lceil (1 - \delta)(n_{\text{cal}} + 1) \rceil$,     $\tau_\delta \leftarrow S_{(k)}$                         ▷ Upper quantile threshold
6: **Inference: Projection-based Scoring and Decision**
7: Encode $\boldsymbol{x}$ and obtain $\boldsymbol{z}$ via reparameterization                      ▷ Latent representation
8: **for** $\boldsymbol{a} \in \mathcal{A}$ **do**
9:     Compute $T(\boldsymbol{z}, \boldsymbol{a})$ in Eq. (5)                            ▷ 1-D normalized deviation
10: **end for**
11: Compute $S(\boldsymbol{x})$ in Eq. (8)                               ▷ Aggregated anomaly score
12: **if** $S(\boldsymbol{x}) \leq \tau_\delta$
13:     $\psi(\boldsymbol{x}) \leftarrow 1$                                        ▷ Majority class
14: **else**
15:     $\psi(\boldsymbol{x}) \leftarrow 2$                                        ▷ Minority class
16: **end if**
17: **return** $\psi(\boldsymbol{x})$ and $S(\boldsymbol{x})$

---

# 3 Experiments

This section provides a comprehensive evaluation of our proposed VAE-Inf on multiple benchmark datasets, focusing on classification performance under severe class imbalance. In addition, we examine the trade-off between Type-I and Type-II errors to evaluate the effectiveness of the proposed error-control mechanism.

## 3.1 Experimental Settings

**Datasets.** We evaluate the proposed VAE-Inf on three data domains: tabular, image, and high-dimensional biomedical data. In the main text, we focus on three tabular datasets (*Credit Card Fraud Detection*, *Backdoor Attack Detection*, and *Census*) and the TCGA Pan-Cancer dataset, which together cover diverse application domains, imbalance levels, and feature dimensions. We report the *minority-class proportion* to characterize the degree of class imbalance, defined as $\rho = \frac{N_2}{N_1+N_2}$, where $N_1$ and $N_2$ denote the numbers of majority- and minority-class samples, respectively. A smaller value of $\rho$ indicates a higher class imbalance level. For TCGA, we use a one-vs-rest protocol in which each cancer type is treated in turn as the minority class. For data preprocessing, we use a 6:2:2 train/validation/test split with preserved class imbalance. Continuous features are standardized using training-set statistics and the same transformation is applied to the validation and test sets. Additional image-domain experiments on MNIST and CIFAR-10, together with detailed dataset descriptions and experimental protocols, are reported in Appendix A.3.

**Methods for Comparison.** We compare our method with five representative deep anomaly detection baselines: DeepSVDD, DeepSAD, DevNet, FeaWAD, and PReNet (Ruff et al., 2018; Pang et al., 2019; Zhou et al., 2022; Pang et al., 2023). These methods are selected to cover the main paradigms of deep anomaly detection under limited supervision; detailed descriptions are provided in Appendix A.1.

**Implementation Details.** All methods are trained under the same clean-label protocol for fair comparison. For implementation, all baselines follow their original papers or standardized open-source implementations from DeepOD and ADBench (Xu et al., 2023; 2024; Han et al., 2022). For our method VAE-inf, the Stage 2 hyperparameters $\alpha$ and $\beta$ are selected on the validation set from a predefined grid and then fixed for test-set evaluation. We use a batch size of 128 and sample 32 random projection directions for projection-based scoring. Additional architectural choices, hyperparameter settings, and training details are given in Appendix A.3.

**Evaluation metrics.** We evaluate model performance using three widely adopted metrics: the Area Under the Receiver Operating Characteristic Curve (AUC-ROC), the Area Under the Precision–Recall Curve (AUC-PR) and F1-score. Since AUC-PR is generally more informative than AUC-ROC under severe imbalance (Saito & Rehmsmeier, 2015), we treat it as a primary threshold-free metric. For F1-score, we adopt a unified thresholding strategy: for each dataset, all methods predict the same fraction of samples as minority-class instances, where this fraction is set to the dataset-specific minority-class proportion. Additional discussion on the choice and interpretation of evaluation metrics is provided in Appendix A.3.

## 3.2 Main Experiment Results

### 3.2.1 Tabular Datasets

Table 1 summarizes the performance of all methods on five tabular datasets with varying degrees of class imbalance. Across these benchmarks, our method achieves highly competitive AUC-ROC and consistently strong AUC-PR and F1-score performance, demonstrating its strength in detecting rare anomalous events. In particular, AUC-PR—the most informative metric under severe imbalance—shows substantial and stable gains over all baselines. For instance, on the *Credit Card* dataset with only 0.17% minority-class samples, our method achieves the highest AUC-PR and F1-score, outperforming strong discriminative approaches such as DeepSAD and PReNet.

To further evaluate robustness under extreme class imbalance, we construct more challenging variants of the *Backdoor* and *Census* datasets by reducing the minority-class proportion in the training set to approxi-

mately 0.20%, while keeping the validation and test sets fixed for evaluation. A clear trend emerges as the training class imbalance becomes more severe. When the available minority-class samples are reduced to this extremely low level, the performance of existing methods decreases substantially, especially in terms of AUC-PR and F1-score. In contrast, our method maintains robust performance and delivers the best results across all metrics in the most imbalanced settings. These improvements suggest that learning a distribution-aware latent representation and using projection-based statistical anomaly scores remain effective even when discriminative baselines become less stable due to the scarcity of minority-class training samples.

In general, the tabular results demonstrate that our VAE-Inf performs on par with or better than state-of-the-art discriminative detectors under moderate imbalance, while offering pronounced gains when the minority-class proportion falls far below 1%.

Table 1: Performance comparison (AUC-ROC / AUC-PR / F1-score, all in percentage). Here, $\rho = N_2/(N_1 + N_2)$ denotes the minority-class proportion. Best and second-best results for each metric per dataset are highlighted in bold and underlined, respectively. All results are averaged over ten independent runs and presented as mean ± standard deviation.

| Metric | DevNet | FeaWAD | DeepSAD | PReNet | DeepSVDD | VAE-inf |
|---|---|---|---|---|---|---|
| **Credit Card** ($\rho = 0.17\%$) | | | | | | |
| AUC-ROC ↑ | 96.06±0.57 | 96.86±0.43 | 95.46±0.80 | **98.03**±0.30 | 86.19±6.82 | 97.48±0.31 |
| AUC-PR ↑ | 45.32±6.93 | 56.85±6.00 | 84.11±1.38 | 70.50±4.92 | 25.95±15.13 | **85.61**±1.45 |
| F1-score ↑ | 51.53±4.26 | 59.59±4.45 | 81.94±1.45 | 71.22±2.57 | 35.40±13.61 | **83.57**±1.87 |
| **Backdoor** ($\rho = 2.44\%$) | | | | | | |
| AUC-ROC ↑ | 98.76±0.44 | 99.09±0.21 | **99.79**±0.10 | 95.74±0.27 | 78.01±7.06 | 99.31±0.13 |
| AUC-PR ↑ | 93.61±0.59 | 91.92±1.85 | **97.61**±0.32 | 89.58±0.48 | 36.88±5.36 | 97.41±0.63 |
| F1-score ↑ | 89.55±0.83 | 88.52±1.75 | **94.14**±0.29 | 86.91±0.35 | 40.70±4.25 | 93.60±1.01 |
| **Backdoor** ($\rho = 0.20\%$) | | | | | | |
| AUC-ROC ↑ | 99.17±0.34 | 97.60±0.88 | 99.24±0.27 | 95.65±0.34 | 86.25±6.50 | **99.26**±0.34 |
| AUC-PR ↑ | 94.14±0.24 | 83.35±3.89 | 95.49±0.80 | 86.66±0.44 | 54.74±8.17 | **96.07**±0.95 |
| F1-score ↑ | 89.96±0.76 | 83.51±2.45 | 91.81±0.47 | 85.71±0.34 | 52.66±9.57 | **93.73**±1.88 |
| **Census** ($\rho = 6.20\%$) | | | | | | |
| AUC-ROC ↑ | 90.49±0.65 | 89.80±0.65 | 90.21±0.37 | 92.66±0.04 | 56.18±6.39 | **93.88**±0.08 |
| AUC-PR ↑ | 39.72±3.38 | 41.27±4.45 | 49.93±0.53 | 54.07±0.19 | 7.42±0.83 | **59.04**±0.39 |
| F1-score ↑ | 43.82±2.54 | 44.99±3.63 | 50.30±0.37 | 52.67±0.17 | 7.61±1.03 | **55.70**±0.42 |
| **Census** ($\rho = 0.21\%$) | | | | | | |
| AUC-ROC ↑ | 88.88±0.50 | 86.76±1.07 | 75.12±1.01 | 88.00±0.47 | 57.44±4.08 | **90.61**±0.28 |
| AUC-PR ↑ | 42.16±0.37 | 30.98±2.23 | 26.43±0.86 | 46.68±0.50 | 7.95±0.61 | **52.39**±1.26 |
| F1-score ↑ | 44.12±0.30 | 37.01±2.23 | 28.37±1.28 | 47.23±0.43 | 9.21±0.95 | **51.08**±1.05 |

### 3.2.2 TCGA Pan-Cancer Results

Table 2 summarizes the performance across 33 one-vs-rest cancer detection tasks on the TCGA pan-cancer dataset. This benchmark is particularly challenging due to its extremely high dimensionality (20,531 genomic features) and highly skewed class distribution. Across all three metrics, our method ranks consistently among the top-performing approaches, achieving the highest AUC-PR (95.58) and F1-score (93.52), while also attaining the second-highest AUC-ROC.

To further evaluate performance under extreme imbalance in realistic rare disease scenarios, we examine five cancer types with prevalence below 1% in the cohort. As shown in Table 2, several baselines struggle in this setting. DeepSAD and PReNet remain competitive, though their performance varies considerably across metrics. PReNet obtains the highest AUC-ROC (99.55) but yields lower AUC-PR, indicating a less favorable precision–recall balance when minority-class samples are exceedingly scarce. In contrast, our method achieves the best AUC-PR (79.66) and F1-score (76.24), and the second-best AUC-ROC, demonstrating improved stability across all three evaluation measures under ultra-imbalanced conditions. These findings highlight that the proposed generative–statistical modeling approach is competitive on average across all 33 tasks and becomes particularly advantageous when minority classes are extremely rare. By explicitly learning the majority-class distribution and grounding anomaly detection in projection-based deviation scoring, our method maintains strong discriminative ability even in high-dimensional, ultra-imbalanced biomedical settings.

Table 2: Performance comparison on TCGA datasets (AUC-ROC / AUC-PR / F1-score, all in percentage). Best and second-best results for each metric per dataset are highlighted in bold and underlined, respectively. Results for TCGA-Full are averaged over all 33 one-vs-rest experiments, while results for TCGA-Rare are averaged over five selected rare-cancer experiments with minority proportion $\rho \leq 1\%$.

| Metric | DevNet | FeaWAD | DeepSAD | PReNet | DeepSVDD | VAE-inf |
|---|---|---|---|---|---|---|
| **TCGA-Full** | | | | | | |
| AUC-ROC ↑ | 99.08±2.32 | 98.60±1.84 | 98.62±3.06 | **99.85**±0.24 | 50.60±7.06 | 99.58±0.99 |
| AUC-PR ↑ | 88.51±20.96 | 82.66±20.35 | 94.53±11.18 | 94.06±12.31 | 3.47±2.33 | **95.58**±8.16 |
| F1-score ↑ | 85.19±21.95 | 79.00±19.39 | 92.51±11.28 | 91.00±14.27 | 3.51±3.50 | **93.52**±9.17 |
| **TCGA-Rare** (five selected types, $\rho \leq 1\%$) | | | | | | |
| AUC-ROC ↑ | 96.35±4.70 | 97.82±1.27 | 93.12±4.86 | **99.55**±0.33 | 55.19±2.91 | 98.32±1.93 |
| AUC-PR ↑ | 63.62±25.16 | 51.83±27.68 | 73.00±15.89 | 73.73±19.74 | 1.82±1.40 | **79.66**±10.35 |
| F1-score ↑ | 60.05±23.72 | 51.82±26.77 | 71.73±15.21 | 67.27±21.75 | 3.61±4.94 | **76.24**±10.05 |

### 3.2.3 Metrics Discussion

While our method is not always the top performer in AUC-ROC, it consistently ranks strongest in AUC-PR and F1-score across nearly all datasets, which are more informative in severely imbalanced settings. As noted in Saito & Rehmsmeier (2015), when the minority-class proportion is extremely small, ROC-based evaluation can be overly influenced by the large number of true negatives. Under such conditions, even large differences in false positives produce only minimal changes in the false-positive rate, causing AUC-ROC to remain high and to be relatively insensitive to improvements in minority-class detection. In contrast, the precision–recall curve directly captures the trade-off between true positives and false positives, making AUC-PR far more informative under severe imbalance. Therefore, the consistently superior AUC-PR and F1 results of our method provide a more reliable indication of practical detection capability, even in cases where AUC-ROC differences are small.

### 3.3 Error Control Results

To evaluate the statistical reliability of the proposed decision rule, we examine how the Type-I and Type-II errors vary with the inference-time threshold $\tau$ applied to the score $S$. After training the model, we keep all network parameters fixed and sweep the inference threshold $\tau$ over 100 uniformly sampled values. For each threshold, we compute the corresponding errors on both the validation and test sets. In Figure 2, solid lines denote Type-I errors and dashed lines denote Type-II errors, with validation and test curves plotted together to assess the stability of the calibration rule across data splits.

For the TCGA dataset, we select Rectum Adenocarcinoma as the minority class, corresponding to an minority-class proportion of 1%. Even in this high-dimensional biomedical setting, the validation and test

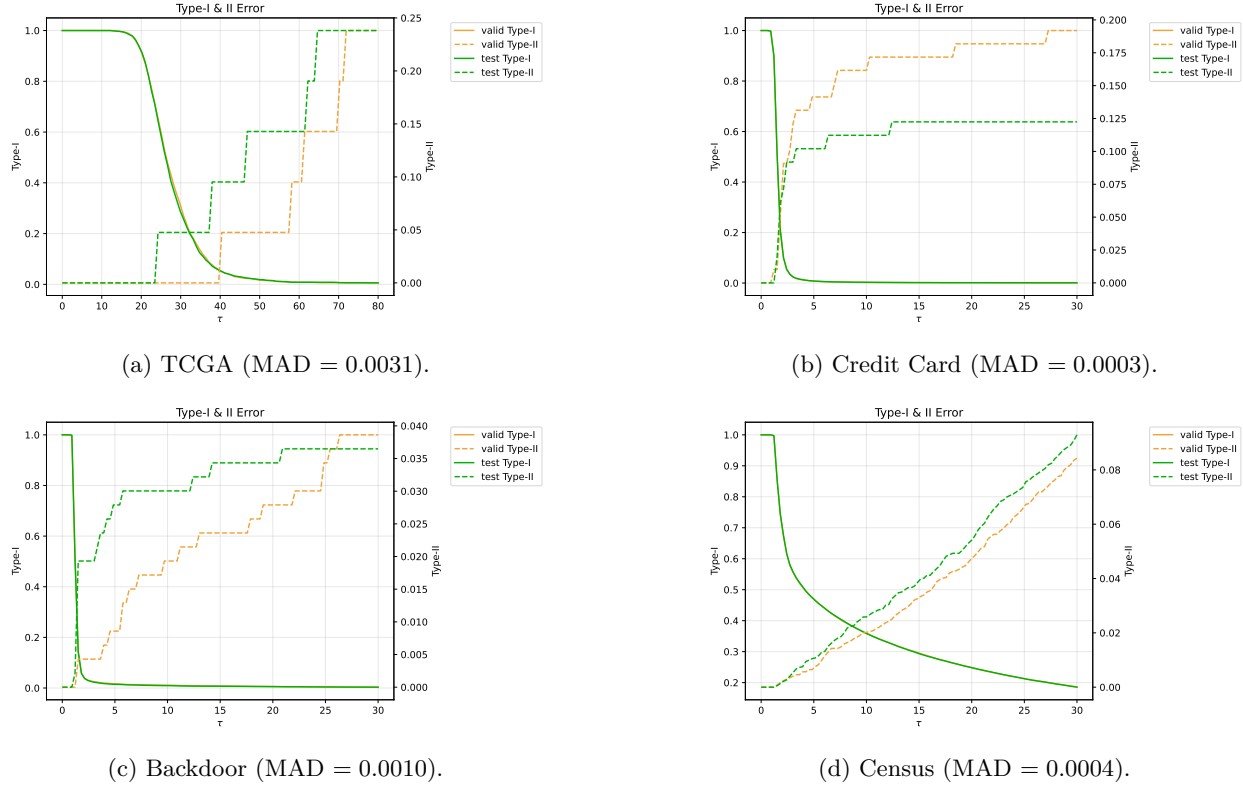

(a) TCGA (MAD = 0.0031).

(b) Credit Card (MAD = 0.0003).

(c) Backdoor (MAD = 0.0010).

(d) Census (MAD = 0.0004).

Figure 2: Type-I (solid) and Type-II (dashed) errors as functions of the inference threshold $\tau$ across datasets. The reported mean absolute deviations (MAD) between the validation and test Type-I curves are uniformly small, indicating that the calibrated decision rule generalizes stably across data splits.

Type-I curves remain closely matched, with a mean absolute deviation of only 0.0031, indicating that the proposed calibration mechanism remains stable beyond tabular benchmarks.

A similar pattern is observed on the tabular datasets. On Credit Card (Figure 2b), the mean absolute deviation between the validation and test Type-I curves is only 0.0003, corresponding to roughly 17 majority samples out of more than 56,000. Similar stability is observed on the Backdoor (Figure 2c) and Census (Figure 2d) datasets. Overall, these results indicate that the proposed score induces stable error profile across validation and test splits, enabling reliable threshold calibration without overfitting to a particular validation split.

Across the entire range of $\tau$, the Type-I curves for different data splits remain closely aligned, indicating that the learned majority reference induces a highly stable decision boundary under threshold variation. By contrast, the Type-II curves exhibit more noticeable variation. This is expected because the minority class is substantially smaller and therefore more sensitive to sampling variability and the stochasticity of fine-tuning. At the same time, increasing $\tau$ enlarges the acceptance region for the majority class, which monotonically reduces the Type-I error but increases the miss-detection rate. Together, these patterns illustrate the fundamental trade-off between statistical conservativeness and anomaly sensitivity.

Table 3 reports the Type-II error under a prescribed Type-I error level ($\delta = 0.01$) and the Type-I error under a prescribed Type-II error level ($\delta = 0.1$), where the corresponding decision threshold is calibrated on the validation set and then evaluated on the test set. For completeness, we also report the test Type-I error at the validation-selected operating point whose validation Type-II is set to 0.1. The proposed method consistently achieves the best or near-best performance across all datasets, demonstrating its effectiveness in meeting target error constraints while maintaining stable generalization under the distribution shift from validation to test data.

Table 3: Type-II (at Type-I = 0.01) and Type-I (at Type-II = 0.1) errors on test set after calibrating the threshold on the validation set. Best results for each metric are highlighted in bold.

| Dataset | Error Type | DeepSVDD | DevNet | FeaWAD | DeepSAD | PReNet | VAE-inf |
|---|---|---|---|---|---|---|---|
| Credit Card (0.17%) | Type-II ↓ | 0.3571 | 0.1939 | 0.1122 | 0.1122 | **0.1020** | **0.1020** |
| | Type-I ↓ | 0.4461 | 0.0595 | 0.1205 | 0.1441 | 0.0683 | **0.0455** |
| Backdoor (2.44%) | Type-II ↓ | 0.8712 | 0.0730 | 0.0901 | **0.0300** | 0.1202 | **0.0300** |
| | Type-I ↓ | 0.8792 | 0.0030 | 0.0052 | **0.0002** | 0.0549 | 0.0003 |
| Backdoor (0.20%) | Type-II ↓ | 0.5021 | 0.0708 | 0.1524 | 0.0708 | 0.1288 | **0.0536** |
| | Type-I ↓ | 0.4312 | 0.0025 | 0.0476 | 0.0008 | 0.0489 | **0.0007** |
| Census (6.20%) | Type-II ↓ | 0.9906 | 0.8810 | 0.8333 | 0.7052 | 0.6877 | **0.6330** |
| | Type-I ↓ | 0.9416 | 0.2502 | 0.1933 | 0.2958 | 0.1740 | **0.1647** |
| Census (0.21%) | Type-II ↓ | 0.9833 | 0.7833 | 0.8883 | 0.8331 | 0.7491 | **0.6828** |
| | Type-I ↓ | 0.6724 | 0.2587 | 0.3289 | 0.5624 | 0.2661 | **0.2494** |
| TCGA (1.00%) | Type-II ↓ | 1.0000 | 0.6190 | 0.5714 | **0.1429** | 0.4286 | **0.1429** |
| | Type-I ↓ | 0.7933 | 0.0333 | 0.0304 | 0.0087 | 0.0251 | **0.0082** |

Overall, these results verify that the proposed inference procedure provides reliable and interpretable error control: for a user-specified target error level $\delta$, the calibration step selects the corresponding decision threshold so as to satisfy the desired constraint. This enables practitioners to specify operational requirements directly in terms of error tolerance, rather than tuning an ad hoc threshold parameter.

## 4 Conclusion

This paper presented VAE-Inf: a two-stage generative framework for imbalanced classification that combines deep representation learning with statistically principled decision-making. By training a VAE solely on majority-class data, the proposed method constructs a reliable latent reference distribution that captures the intrinsic structure of normal samples without contamination from scarce minority data. A distribution-aware fine-tuning stage then exploits limited minority information to enhance separability in latent space through variance-normalized projection statistics.

A key advantage of the proposed approach lies in its statistical interpretability. The projection-based score naturally corresponds to a hypothesis testing statistic under the learned majority distribution, enabling a transparent acceptance–rejection mechanism. Moreover, the introduction of an empirical quantile calibration procedure provides exact finite-sample control of the Type-I error under minimal exchangeability assumptions, independent of the correctness of the learned generative model. This distinguishes our method from existing imbalanced learning approaches that rely on heuristic thresholds or asymptotic approximations.

Experiment results across diverse datasets demonstrate that the proposed VAE-Inf achieves superior classification performance for highly imbalanced data while offering explicit Type-I error guarantees. Beyond imbalanced classification, the methodology developed in this work suggests a broader perspective on integrating deep generative models with distribution-free statistical inference. Future work may explore extensions to multi-class imbalance, adaptive projection strategies, and alternative generative priors, as well as applications to other anomaly-sensitive domains such as biomedical analysis and safety-critical systems.

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

# A  Appendix

## A.1  Related Work

Deep anomaly detection (AD) under limited supervision has received increasing attention in recent years. In practice, some labeled data is often available and should be fully exploited. Leveraging such labels effectively to learn expressive representations of normality and abnormality is therefore essential for accurate anomaly detection (Pang et al., 2021). Existing methods exploit this weak supervision in different ways, which can be broadly grouped into geometric compactness modeling, statistical score learning, reconstruction-guided representation learning, and relational supervision.

A representative line of work focuses on geometric compactness of normal data in latent space. Deep-SAD (Ruff et al., 2020) extends Deep SVDD (Ruff et al., 2018) to the semi-supervised setting by incorporating labeled normal and anomalous samples into a unified objective. Unlabeled and normal samples are

encouraged to concentrate around a latent center, while labeled anomalies are pushed away, yielding low-entropy representations for normal data. This framework effectively leverages limited supervision to improve robustness over purely unsupervised approaches.

Another line of work formulates AD as direct statistical score learning. DevNet (Pang et al., 2019) proposes to optimize anomaly scores against a predefined Gaussian reference distribution using a deviation loss. Normal samples are constrained to stay near the reference mean, while labeled anomalies are encouraged to deviate significantly, resulting in statistically interpretable decision thresholds. This design enables efficient training and scalability, and provides an explicit connection between anomaly scores and significance levels.

To better capture the intrinsic structure of normal data, recent methods integrate reconstruction-based representation learning with discriminative objectives. FeaWAD (Zhou et al., 2022) employs an autoencoder to model the normal data manifold and represents each sample using a three-factor feature tuple $(h, r, e)$, corresponding to latent embedding, reconstruction residual, and reconstruction error magnitude. These features are subsequently fed into a discriminative scoring network.

More recently, relational supervision has been proposed to further amplify weak supervision. PReNet (Pang et al., 2023) constructs pairwise relationships among samples (normal–normal, normal–anomaly, anomaly–anomaly) and learns to predict their relative anomaly ordering. This pairwise formulation significantly enlarges the effective supervision signal and improves generalization to unseen anomalies.

In summary, existing AD methods leverage limited labels through diverse mechanisms, ranging from geometric constraints and reference-based scoring to manifold-aware representations and relational learning. While effective, most approaches either rely on simplified assumptions about the normal distribution or produce heuristic anomaly scores without an explicit, data-driven statistical decision rule. These limitations motivate approaches that jointly learn faithful majority-class representations and enable interpretable, statistically grounded inference under severe class imbalance.

### A.2 Finite-Sample Guarantee via Empirical Quantile Calibration

In this section, we provide a rigorous proof for the Type-I error control achieved by the calibration scheme. Theorem 1 establishes a finite-sample guarantee under the sole assumption of exchangeability of the scores.

*Proof.* If $k = n_{\text{cal}} + 1$, then by definition $\tau_\delta = +\infty$, and therefore

$$\mathbb{P}\big(S_{n_{\text{cal}}+1} \leq \tau_\delta\big) = 1 \geq 1 - \delta.$$

Thus, it remains to consider the case $k \leq n_{\text{cal}}$, for which $\tau_\delta = S_{(k)}$.

By exchangeability and the assumption that the scores are almost surely distinct, the rank of $S_{n_{\text{cal}}+1}$ among

$$\{S_1, \ldots, S_{n_{\text{cal}}}, S_{n_{\text{cal}}+1}\}$$

is uniformly distributed over $\{1, \ldots, n_{\text{cal}} + 1\}$. That is, for every $j \in \{1, \ldots, n_{\text{cal}} + 1\}$,

$$\mathbb{P}\big(\operatorname{rank}(S_{n_{\text{cal}}+1}) = j\big) = \frac{1}{n_{\text{cal}} + 1}.$$

Since $\tau_\delta = S_{(k)}$, the event $S_{n_{\text{cal}}+1} \leq \tau_\delta$ is equivalent to the event that the rank of $S_{n_{\text{cal}}+1}$ is at most $k$. Hence

$$\mathbb{P}\big(S_{n_{\text{cal}}+1} \leq \tau_\delta\big) = \mathbb{P}\big(\operatorname{rank}(S_{n_{\text{cal}}+1}) \leq k\big) = \frac{k}{n_{\text{cal}} + 1}.$$

Substituting

$$k = \lceil (1 - \delta)(n_{\text{cal}} + 1) \rceil$$

and the elementary inequality $\lceil x \rceil \geq x$, we obtain

$$\frac{k}{n_{\text{cal}} + 1} \geq \frac{(1 - \delta)(n_{\text{cal}} + 1)}{n_{\text{cal}} + 1} = 1 - \delta.$$

Therefore,

$$\mathbb{P}\big(S_{n_{\text{cal}}+1} \leq \tau_\delta\big) \geq 1 - \delta.$$

This completes the proof. □

## A.3 Experimental Setting

### A.3.1 Dataset Description

**Tabular datasets.** We consider three representative tabular imbalance classification datasets that cover financial risk assessment, network intrusion detection, and socioeconomic analysis. The *Credit Card Fraud Detection* dataset from Kaggle exhibits an extreme minority-class proportion of 0.17%, reflecting realistic fraud detection scenarios. The *Backdoor Attack Detection* dataset is derived from the UNSW-NB15 benchmark (Moustafa & Slay, 2015) and contains 2.44% anomalous traffic instances, representing a modern network security challenge. The *Census* dataset is constructed from the U.S. Census Bureau database, where the task is to identify individuals with income exceeding 50K dollars per year; only 6.2% of the population belongs to this high-income group.

These datasets span diverse domains and imbalance levels, providing a broad evaluation of our method. A summary of dataset description is presented in Table 4.

Table 4: Overview of the tabular datasets. The minority-class proportion is defined as $\rho = N_2/(N_1 + N_2)$.

| Dataset | #Samples | Features | #Minority Samples | Minority Prop. | Category |
|---|---|---|---|---|---|
| Credit Card | 284,807 | 30 | 492 | 0.17% | Finance |
| Backdoor | 95,329 | 196 | 2,329 | 2.44% | Network |
| Census | 299,285 | 500 | 18,568 | 6.20% | Sociology |

All tabular features are standardized to zero mean and unit variance. For consistency across benchmarks, each dataset is partitioned into training, validation, and test splits with a ratio of 6:2:2, and the minority-class proportion is preserved within each split.

**Image datasets.** Following the experimental protocol of DeepSAD (Ruff et al., 2020), we evaluate our method on two benchmark image datasets: MNIST, and CIFAR-10. In the *no-pollution* setting, one class is designated as the majority class, while one of the remaining nine classes is randomly selected as the minority class. This procedure is repeated for all 10 majority classes and all 9 minority classes, resulting in a total of 90 experiments per dataset.

To simulate realistic extreme imbalance scenarios, the fraction of labeled training anomalies is set to $\rho = \frac{N_2}{N_1 + N_2} = 0.005$. The same sampling ratio is applied consistently to both the training and test sets. Performance is reported as the average values across all 90 runs.

**TCGA Pan-Cancer dataset.** We further evaluate our method on a large-scale biomedical dataset: the TCGA Pan-Cancer cohort obtained from the UCSC Xena platform (`http://xena.ucsc.edu`). We use the `genomicMatrix` representation, where each row corresponds to a patient sample and each column corresponds to a genomic feature (molecular identifier). The dataset consists of 10,459 samples and 20,531 genomic features across 33 distinct cancer types. The class distribution is highly imbalanced, as shown in Table 5.

The imbalance is substantial: the most common subtype (*breast invasive carcinoma*) constitutes over 11% of the dataset, whereas several rare malignancies, such as *cholangiocarcinoma* and *Mesothelioma*, each account for fewer than 1% of samples. This makes TCGA a highly challenging and biologically meaningful benchmark. For evaluation, we adopt a one-vs-rest protocol. Each of the 33 cancer types is treated in turn as the minority class, while samples from the remaining 32 types form the majority class. This yields 33 separate experiments. In addition to reporting the average performance over all 33 one-vs-rest tasks, we also report results on a

Table 5: Distribution of phenotype classes in the TCGA pan-cancer dataset.

| Class | Occurrences | Proportion |
|---|---|---|
| Breast invasive carcinoma | 1218 | 11.65% |
| Kidney clear cell carcinoma | 606 | 5.79% |
| Lung adenocarcinoma | 576 | 5.51% |
| Thyroid carcinoma | 572 | 5.47% |
| Head & neck squamous cell carcinoma | 566 | 5.41% |
| Lung squamous cell carcinoma | 553 | 5.29% |
| Prostate adenocarcinoma | 550 | 5.26% |
| Brain lower grade glioma | 530 | 5.07% |
| Skin cutaneous melanoma | 474 | 4.53% |
| Stomach adenocarcinoma | 450 | 4.30% |
| Bladder urothelial carcinoma | 426 | 4.07% |
| Liver hepatocellular carcinoma | 423 | 4.04% |
| Colon adenocarcinoma | 329 | 3.15% |
| Kidney papillary cell carcinoma | 323 | 3.09% |
| Cervical & endocervical cancer | 308 | 2.94% |
| Ovarian serous cystadenocarcinoma | 308 | 2.94% |
| Sarcoma | 265 | 2.53% |
| Uterine corpus endometrioid carcinoma | 201 | 1.92% |
| Esophageal carcinoma | 196 | 1.87% |
| Pheochromocytoma & paraganglioma | 187 | 1.79% |
| Pancreatic adenocarcinoma | 183 | 1.75% |
| Acute myeloid leukemia | 173 | 1.65% |
| Glioblastoma multiforme | 172 | 1.64% |
| Testicular germ cell tumor | 156 | 1.49% |
| Thymoma | 122 | 1.17% |
| Rectum adenocarcinoma | 105 | 1.00% |
| Kidney chromophobe | 91 | 0.87% |
| Mesothelioma | 87 | 0.83% |
| Uveal melanoma | 80 | 0.76% |
| Adrenocortical cancer | 79 | 0.76% |
| Uterine carcinosarcoma | 57 | 0.54% |
| Diffuse large B-cell lymphoma | 48 | 0.46% |
| Cholangiocarcinoma | 45 | 0.43% |

selected rare-cancer subset with minority proportion $\rho \leq 1\%$, consisting of *rectum adenocarcinoma*, *uterine carcinosarcoma*, *kidney chromophobe*, *cholangiocarcinoma*, and *adrenocortical cancer*.

### A.3.2 Implementation Details

All baseline methods are implemented following the hyperparameters and model configurations reported in their original papers. For consistency and reproducibility, we adopt the codes from original papers and open-source implementations from DeepOD (Xu et al., 2023; 2024) and ADBench (Han et al., 2022), which provide standardized training pipelines and widely validated configurations for deep anomaly detection.

Our approach employs a vanilla VAE backbone in Stage 1, with the architecture adapted to the structure of each data domain. For tabular datasets, we use a lightweight fully connected encoder–decoder, where both the encoder and decoder contain two hidden layers. For image datasets, MNIST uses a convolutional encoder–

decoder with residual blocks, whereas CIFAR-10 adopts a deeper residual convolutional architecture. For the TCGA Pan-Cancer dataset, which contains over 20,000 genomic features, we use a high-capacity MLP-based VAE with residual feedforward blocks. During Stage 2, the model is fine-tuned using the proposed projection-based regularization. The hyperparameters $\alpha$ and $\beta$ are selected on the validation set from a predefined grid and then fixed for test-set evaluation. The latent dimension $d_l$ and the hyperparameters $\alpha$ and $\beta$ are summarized in Table 6.

Table 6: Hyperparameter settings used in the experiments, including the latent dimension $d_l$ and the validation-selected Stage 2 hyperparameters $\alpha$ and $\beta$.

| Dataset | $d_l$ | $\alpha$ | $\beta$ |
|---|---|---|---|
| Credit Card | 16 | 16 | 2 |
| Backdoor | 20 | 25 | 16 |
| Census | 50 | 25 | 10 |
| MNIST | 16 | 16 | 10 |
| CIFAR-10 | 64 | 16 | 10 |
| TCGA Pan-Cancer | 128 | 16 | 2 |

For all experiments, we set the batch size to 128. The numbers of training epochs in Stage 1 and Stage 2 are set to $E_1 = 200$ and $E_2 = 100$, respectively. The corresponding learning rates are $1 \times 10^{-4}$ for Stage 1 and $2 \times 10^{-3}$ for Stage 2. For projection-based scoring, we sample 32 random projection vectors to estimate the anomaly score. All models are optimized using Adam with default momentum parameters, and early stopping is applied based on the validation loss to prevent overfitting.

### A.3.3 Evaluation Metrics

We evaluate model performance using three widely adopted metrics: the Area Under the Receiver Operating Characteristic Curve, the Area Under the Precision–Recall Curve (AUC-PR) and F1-score.

AUC-ROC measures the trade-off between true positive and false positive rates across all decision thresholds. Although widely used, AUC-ROC can be misleading under severe class imbalance, as its value may be dominated by the abundant normal samples rather than the minority-class samples. In contrast, AUC-PR focuses explicitly on the performance of the positive (minority) class by measuring precision and recall. As demonstrated in Saito & Rehmsmeier (2015), AUC-PR is a more informative indicator than AUC-ROC in highly imbalanced settings, where even a large improvement in identifying minority-class samples may produce only marginal changes in AUC-ROC. We compute AUC-PR following the standard average-precision formulation in Schütze et al. (2008).

Finally, we report the F1-score as a threshold-dependent metric that balances precision and recall. Since most baseline methods do not provide explicit decision thresholds, we adopt a unified strategy based on the minority-class proportion reported in Table 4 for threshold selection. For example, in Credit Card Fraud Detection dataset, where the minority-class proportion is 0.17%, the top 0.17% of samples ranked by anomaly scores are predicted as minority class. This strategy ensures a consistent and fair comparison on classification performance across methods by fixing the predicted minority proportion.

### A.4 Ablation and Sensitivity Analysis

### A.4.1 Effect of Stage-2 fine-tuning

Table 7 reports the performance of the proposed VAE-Inf before and after Stage 2 on tabular datasets. A consistent and substantial improvement is observed in all evaluation metrics. In particular, Stage 1 alone yields limited discriminative performance, especially on highly imbalanced datasets. This is consistent with the observation in the latent space, where minority samples are not well separated from the majority manifold.

After incorporating Stage 2, all metrics improve significantly, demonstrating that Stage 2 effectively enhances the discriminative capability of the learned representation.

Table 7: Performance comparison between Stage 1 and Stage 2 (AUC-ROC / AUC-PR / F1-score, all in percentage). Best results for each metric are highlighted in bold. All results are averaged over ten independent runs and presented as mean ± standard deviation.

| Method | AUC-ROC ↑ | AUC-PR ↑ | F1-score ↑ |
|---|---|---|---|
| **Credit Card** ($\rho = 0.17\%$) | | | |
| Stage 1 | 87.51±1.48 | 4.30±0.58 | 8.27±1.25 |
| Stage 1&2 | **97.48**±0.31 | **85.61**±1.45 | **83.57**±1.87 |
| **Backdoor** ($\rho = 2.44\%$) | | | |
| Stage 1 | 77.89±2.71 | 29.30±2.17 | 41.52±1.45 |
| Stage 1&2 | **99.31**±0.13 | **97.41**±0.63 | **93.60**±1.01 |
| **Census** ($\rho = 6.20\%$) | | | |
| Stage 1 | 42.80±1.72 | 5.51±0.17 | 5.37±0.27 |
| Stage 1&2 | **93.88**±0.08 | **59.04**±0.39 | **55.70**±0.42 |

### A.4.2 Sensitivity to Hyperparameters

We examine the influence of the hyperparameters $\alpha$ and $\beta$ in the proposed distribution-aware regularization term Eq. (7). The parameter $\alpha$ controls the extent of the latent confidence region of the majority distribution, while $\beta$ regulates the strength with which minority embeddings are encouraged to lie outside this region. The larger $\alpha$ allows for a wider admissible range for majority samples, whereas the larger $\beta$ increases the penalty for minority embeddings that remain close to the majority reference region.

Figure 3 presents a sensitivity analysis of AUC-PR with respect to $\beta$ under different choices of $\alpha$ on the Credit Card dataset. Within the considered grid, the most favorable performance on this dataset is observed at ($\alpha = 16, \beta = 2$). The results show that the performance is sensitive to the balance between majority tolerance and minority separation. Extremely small $\alpha$ (overly restrictive boundaries) or excessively large $\beta$ (over-separation) both lead to performance degradation, indicating that balanced regularization is essential for stable discrimination under high imbalance. The selected values of $\alpha$ and $\beta$ vary across datasets and are determined through validation set selection in the main experiments.

### A.5 Image Datasets Results

Table 8 reports results on MNIST and CIFAR-10 under the $\rho = 0.005$ protocol. Since most baselines are designed for tabular inputs, we compare our method with DeepSAD, the only baseline offering official support for image anomaly detection. All results are averaged over 90 experiments spanning all normal–anomaly class combinations.

Across both datasets, our method achieves competitive AUC-ROC and consistently stronger AUC-PR and F1-score performance. On MNIST, where anomaly structure is relatively simple, both methods obtain similar AUC-ROC, but our approach yields noticeably higher AUC-PR (+6.6 points on average) and F1-score, indicating more precise identification of rare minority-class samples. The advantage becomes larger on CIFAR-10 dataset. While DeepSAD attains a slightly higher AUC-ROC, its AUC-PR drops sharply due to poor precision under extreme imbalance. In contrast, our distribution-aware scoring achieves a large improvement in AUC-PR (from 18.53 to 62.07) and F1-score, reflecting a significantly enhanced ability to rank and classify anomalous image samples. Overall, the image results demonstrate that the proposed VAE-Inf remains effective beyond tabular domains.

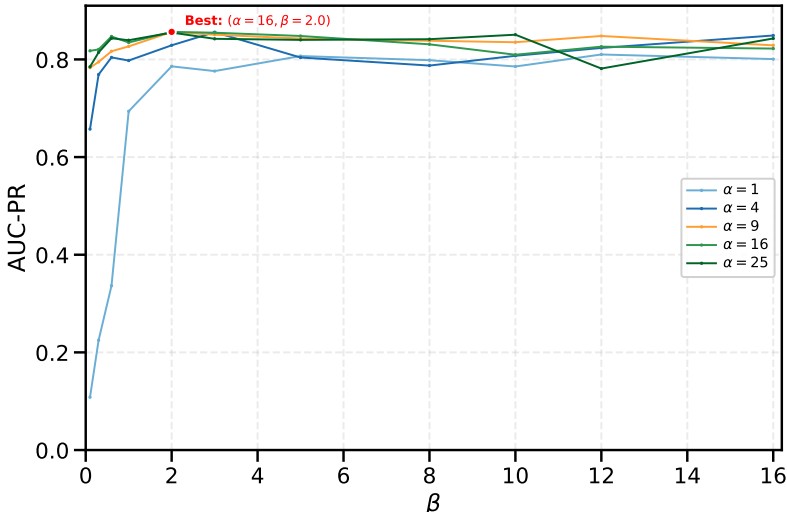

Figure 3: Sensitivity of AUC-PR to $\beta$ under different choices of $\alpha$ on the Credit Card dataset. Optimal performance occurs at $(\alpha = 16, \beta = 2)$.

Table 8: Comparison of AUC-ROC and AUC-PR on image datasets ( $\rho = 0.005$ ). Results are averaged over 90 experiments.

| Method | AUC-ROC ↑ | AUC-PR ↑ | F1-score ↑ |
|---|---|---|---|
| **MNIST** | | | |
| DeepSAD | 95.22±3.61 | 67.36±18.20 | 62.59±16.38 |
| VAE-inf (Ours) | **95.25±3.43** | **73.94±14.38** | **68.04±13.74** |
| **CIFAR-10** | | | |
| DeepSAD | **72.54±8.35** | 18.53±10.18 | 21.31±10.06 |
| VAE-inf (Ours) | 71.46±7.62 | **62.07±11.60** | **55.55±9.24** |

## A.6 Additional Error-Control and Calibration Results

### A.6.1 Stability across Training, Validation, and Test Splits

We evaluate the stability of the proposed decision rule across the training, validation, and test splits of the Credit Card dataset, using hyperparameters $\alpha = 16$ and $\beta = 2$. The three subsets preserve the same minority-class proportion (approximately 0.17%), enabling a controlled examination of error behaviors. Sample statistics are summarized below:

- **Training:** 170,883 samples, 295 minority (0.1726%)

- **Validation:** 56,962 samples, 99 minority (0.1738%)

- **Test:** 56,962 samples, 98 minority (0.1720%)

Table 9 reports the quantitative results obtained by using the same threshold parameter $\tau = 16$ during training and directly deploying it for inference on three sets. A notable observation is the remarkable stability of the Type-I error, which remains within the narrow range of 0.11%–0.16% across all splits. This confirms that the latent boundary learned in Stage 1 is well calibrated: normal samples consistently fall within the estimated majority region, and the decision threshold based on statistical score preserves its

validity across independent partitions. Type-II error exhibits larger variability, increasing from 6.44% in the training set to 17.17% on the validation split, and decreasing again to 12.24% on the test set. This fluctuation is expected due to the limited number of minority samples and the stochasticity inherent in fine-tuning.

Table 9: Confusion matrices for training, validation, and test sets. Rows correspond to true labels and columns to predicted labels.

| True\Pred. | Maj. | Min. |
|---|---|---|
| Maj. | **170,393** | 195 |
| Min. | 19 | **276** |

(a) Training set

| True\Pred. | Maj. | Min. |
|---|---|---|
| Maj. | **56,785** | 78 |
| Min. | 17 | **82** |

(b) Validation set

| True\Pred. | Maj. | Min. |
|---|---|---|
| Maj. | **56,772** | 92 |
| Min. | 12 | **86** |

(c) Test set

### A.6.2 Calibrated Type-I and Type-II Error Control

To further assess the practical utility of empirical calibration, we select $\tau$ on the validation set to achieve a target Type-I error level of 0.01 and evaluate the resulting Type-II error on both validation and test sets. As shown in Table 10, the calibrated test-set Type-I errors remain tightly concentrated around the desired level across all datasets, confirming that the rule provides effective false-positive control. Likewise, the corresponding Type-II errors on the test set closely match those on the validation split. A complementary analysis using a target Type-II level of 0.1 in Table 11 shows the same pattern. Since the empirical Type-II error is discrete for finite minority samples, exact matching to 0.1 is generally impossible. We select the threshold that yields the closest attainable validation Type-II error to 0.1. The resulting test-set errors remain closely aligned with the validation-set calibrations, further highlighting the robustness of the projection-based score.

Table 10: Type-I and Type-II errors after calibrating the threshold on the validation set to achieve Type-I = 0.01. Results are reported for both the validation and test sets across all datasets. Numbers in parentheses indicate the corresponding error count divided by the relevant class size.

| Dataset | Split | Type-I Error ↓ | Type-II Error ↓ |
|---|---|---|---|
| Credit Card (0.17%) | Val | **0.0100** (568/56863) | 0.1313 (13/99) |
| | Test | 0.0107 (607/56864) | 0.1020 (10/98) |
| Backdoor (2.44%) | Val | **0.0100** (186/18600) | 0.0193 (9/466) |
| | Test | 0.0084 (156/18600) | 0.0300(14/466) |
| Backdoor (0.20%) | Val | **0.0100** (186/18600) | 0.0536(25/466) |
| | Test | 0.0084 (156/18600) | 0.0536 (25/466) |
| Census (6.20%) | Val | **0.0100** (561/56144) | 0.6154 (2285/3713) |
| | Test | 0.0102 (572/56143) | 0.6330 (2351/3714) |
| Census (0.21%) | Val | **0.0100** (561/56144) | 0.6601 (2451/3713) |
| | Test | 0.0103 (577/56143) | 0.6828 (2536/3714) |

Table 11: Type-I and Type-II errors after calibrating the threshold on the validation set to obtain the closest attainable empirical Type-II error to 0.1. Results are reported for both the validation and test sets. Numbers in parentheses indicate the corresponding error count divided by the relevant class size.

| Dataset | Split | Type-I Error ↓ | Type-II Error ↓ |
|---|---|---|---|
| Credit Card (0.17%) | Val | 0.0455(2586/56863) | **0.0909**(9/99) |
| | Test | 0.0460(2617/56864) | 0.0918(9/98) |
| Backdoor (2.44%) | Val | 0.0001(2/18600) | **0.0987**(46/466) |
| | Test | 0.0003(5/18600) | 0.0987(46/466) |
| Backdoor (0.20%) | Val | 0.0007(13/18600) | **0.0987**(46/466) |
| | Test | 0.0007(13/18600) | 0.0901(42/466) |
| Census (6.20%) | Val | 0.1650(9264/56144) | **0.0999**(371/3713) |
| | Test | 0.1647(9246/56143) | 0.1061(394/3714) |
| Census (0.21%) | Val | 0.2506(14069/56144) | **0.0999**(371/3713) |
| | Test | 0.2494(14002/56143) | 0.1010(375/3714) |

