# OpenReview forum: "VAE-Inf: A statistically interpretable generative paradigm for imbalanced classification"
_TMLR — Under review for TMLR_

### Review · Reviewer_CCRY · 2026-06-27

**Summary Of Contributions:**

This paper proposes VAE-Inf, a two-stage framework that integrates deep representation learning with statistically interpretable hypothesis testing, to tackle the imbalanced classification task.
In particular, this paper recognizes that under extreme class imbalance, defining what the minority class "is not" is significantly easier than defining what it "is".
In this light, this paper first trains a variational autoencoder exclusively on the majority-class data to construct a reference distribution for the majority class. Then it uses limited minority samples via a distribution-aware fine-tuning process, thereby enhancing class separability while preserving the learned majority structure.
Extensive experiments also demonstrate that VAE-Inf indeed achieves superior classification performance against various baselines.

**Additional Comments:**

None

**Audience:**

Yes

**Audience Explanation:**

Imbalanced classification has been a pervasive challenge within the machine learning community, playing a critical role in vital real-world applications such as pattern recognition, object detection, and medical diagnosis, where misclassifying rare samples can lead to severe or even fatal consequences.
Given this context, TMLR’s audience will be highly interested in this paper, as the proposed VAE-Inf framework elegantly bridges the gap between generative modeling and discriminative classification, offering a robust solution with superior classification performance and reliable Type-I/II error control for imbalanced classification.

**Claims And Evidence:**

Yes

**Claims Explanation:**

1. The authors observe that under extreme imbalance, defining what the minority class "is not" is significantly easier than defining what it "is". This is highly justified because when rare samples are too scarce to learn a reliable distribution, profiling the abundant majority class provides a much more robust reference.
2. Drawing inspiration from anomaly detection, the authors introduce VAE-Inf, which first captures the majority-class reference distribution and then refines the latent space using a distribution-aware loss with limited minority samples.
3. Extensive experiments on several important datasets demonstrate that VAE-Inf indeed achieves superior classification performance while maintaining reliable Type-I/II error control compared with various baselines.

**Requested Changes:**

1. Since the current experiments are primarily formulated for binary classification, extending VAE-Inf to multi-class imbalanced datasets would further broaden its practical impact.
2. It's also necessary to include conventional imbalanced learning baselines (e.g., Focal Loss and LDAM) to validate the superiority of the proposed method.

---

### Review · Reviewer_v2V8 · 2026-07-04

**Summary Of Contributions:**

The paper proposes VAE-Inf, a two-stage method for extreme imbalanced classification that reframes the task as one-class modeling of the majority plus calibrated deviation testing, rather than trying to model a scarce minority directly.In Stage 1, a VAE is trained only on majority-class data, and the per-sample latent posteriors are aggregated into a single reference Gaussian using the closed-form 2-Wasserstein barycenter, which here amounts to averaging the posterior means and averaging the posterior standard deviations. In Stage 2, the encoder is fine-tuned with a hinge-type loss defined over random projection directions, using a variance-normalized projection statistic that keeps majority samples inside a margin and pushes minority samples outside it. At inference, the per-direction statistics are averaged into a single anomaly score S(x), and the decision threshold is set by the empirical upper quantile of majority calibration scores. This yields distribution-free, finite-sample control of the Type-I (false-positive) error under exchangeability, formalized in Theorem 1.

**Strengths**

* The conformal Type-I guarantee is correctly derived (Appendix A.2 is a standard split-conformal argument) and holds independently of whether the generative model is accurate, which the paper states explicitly.

* The evaluation covers a wide range of imbalance levels (minority proportion from 6.20% down to 0.17%) and domains (finance, network security, high-dimensional genomics), and the emphasis on AUC-PR over AUC-ROC is well justified for severe imbalance.

**Weaknesses**

* The Gaussian-reference variance in Eq. (4) provably underestimates the true dispersion of the majority latents, so the chi-square and c-sigma interpretation does not hold on real data; the error control comes from calibration, not from the generative model.

* Superiority is claimed against imbalanced-learning methods (contribution 3), but every baseline actually tested is an anomaly-detection method.

**Additional Comments:**

N/A

**Audience:**

Yes

**Audience Explanation:**

Distribution-free Type-I control layered on a learned latent score, validated across finance, network security, and high-dimensional genomics, is of clear interest to TMLR readers regardless of state-of-the-art status.

**Broader Impact Concerns:**

No major concerns.

**Claims And Evidence:**

No

**Claims Explanation:**

The finite-sample Type-I control (Theorem 1) is correctly proven, empirically confirmed, and holds regardless of the generative model, which is the paper's strongest element. Two claims, however, are not supported by the current evidence.

First, the "statistically interpretable" framing. The reference covariance in Eq. (4) is the squared mean of the posterior standard deviations. But the actual per-dimension dispersion of the majority latents, which are drawn by reparameterization at inference, is the mean of the posterior variances plus the between-sample variance of the posterior means. By Jensen's inequality the first term alone already exceeds the reference value, and the between-sample mean-scatter term is dropped entirely. The reference covariance therefore strictly underestimates the true spread, so the standardized projection is not standard normal on real majority data and the projection statistic does not follow the claimed chi-square distribution. The 3-sigma / 0.27% reading in Section 2.2.2 does not hold, and the interpretability is supplied by the calibration step rather than the Gaussian model.

Second, "superior classification performance" is claimed relative to the imbalanced-learning literature the introduction motivates against (resampling, cost-sensitive, ensemble, generative oversampling, focal, LDAM), but none of these appear as baselines. The claim and the experiments are directly mismatched.

**Requested Changes:**

**Critical**

1. Add imbalanced-classification baselines, such as a class-weighted, focal, or LDAM-margin classifier, and SMOTE combined with a classifier. Alternatively, narrow the superiority claim to weakly-supervised anomaly detection so that it matches the baselines actually used.

2. Please clarify the Gaussian-reference interpretation. Since the reference covariance underestimates the true dispersion of the majority latents, the chi-square interpretation may not hold on real data. It would help to either recompute the reference after Stage 2 using the moment-matched aggregate posterior (the mean of the posterior variances plus the variance of the posterior means), or to show empirically that the projection statistic matches its claimed chi-square distribution on held-out majority data.

3. Ablate the generative component by replacing the VAE with a deterministic autoencoder or a plain encoder while keeping the Stage-2 loss and the calibration. Table 7 shows that Stage 1 alone is nearly non-discriminative, but it does not isolate whether the generative model contributes beyond a plain encoder.

**Strengthening**

4. Ablate the random-projection score against a Mahalanobis-distance score, and report sensitivity to the number of projection directions.

5. Unify the notation across the paper, as the latent dimension appears under several different symbols.

---

### Review · Reviewer_n2kx · 2026-07-13

**Summary Of Contributions:**

The paper studies the imbalanced classification. The authors propose VAE-inf, which adopts a two-stage pipeline to first pretrain a VAE on the majority class and then tune the VAE encoder to improve the class separation. The inference process of VAE-inf uses an averaged projection statistic and an empirical-quantile calibration on held-out majority data to control Type-I error with a standard exchangeability guarantee. Experiments on tabular, image, and TCGA gene-expression datasets show strong AUC-PR and F1 performance and demonstrate controllable Type-I error.

**Audience:**

Yes

**Audience Explanation:**

The studied problem is important to the community.

**Claims And Evidence:**

Yes

**Claims Explanation:**

The authors provide both theoretical and empirical evidence to justify the proposed method.

**Requested Changes:**

1. The error-control guarantee seems to be a natural benefit of projection-score calibration. How is the calibration proceeded with the baselines?

2. Could you discuss the scenario when the number of samples from the minority class is not sufficiently large for statistical tests?

3. In experiments, lots of standard imbalanced learning methods discussed in the introduction seem not to be included or compared.

4. In Table 8, why is the AUC-PR of VAE-inf three times larger than that of DeepSAD, while with a similar AUC-ROC?